



**Sensitivity and identifiability of hydraulic and geophysical parameters from**
**streaming potential signals in unsaturated porous media**
Anis Younes[1,2,3], Jabran Zaouali[1], Francois Lehmann[1], Marwan Fahs[*,1]
[1]LHyGES, Université de Strasbourg/EOST/ENGEES, CNRS, 1 rue Blessig, 67084 Strasbourg, France.
[2]IRD UMR LISAH, F-92761 Montpellier, France
[3]LMHE, ENIT, Tunis, Tunisie
* Contact person: Marwan Fahs
E-mail: fahs@unistra.fr



*Abstract*
Fluid flow in a charged porous medium generates electric potentials called Streaming
potential (SP). The SP signal is related to both hydraulic and electrical properties of the soil.
In this work, Global Sensitivity Analysis (GSA) and parameter estimation procedures are
performed to assess the influence of hydraulic and geophysical parameters on the SP signals
and to investigate the identifiability of these parameters from SP measurements. Both
procedures are applied to a synthetic column experiment involving a falling head infiltration
phase followed by a drainage phase.
GSA is used through variance-based sensitivity indices, calculated using sparse Polynomial
Chaos Expansion (PCE). To allow high PCE orders, we use an efficient sparse PCE algorithm
which selects the best sparse PCE from a given data set using the Kashyap Information
Criterion (KIC). Parameter identifiability is performed using two approaches: the Bayesian
approach based on the Markov Chain Monte Carlo (MCMC) method and the First-Order
Approximation (FOA) approach based on the Levenberg Marquardt algorithm.
GSA results show that at short times, the saturated hydraulic conductivity $(K_s)$ and the
voltage coupling coefficient at saturation $(C_{sat})$ are the most influential parameters, whereas,
at long times, the residual water content $(\theta_r)$, the Mualem-van Genuchten parameter $(n)$ and
the Archies's saturation exponent $(n_a)$ become influential with strong interactions between
them. The Mualem-van Genuchten parameter $(\alpha)$ has a very weak influence on the SP
signals during the whole experiment.
Results of parameter estimation show that, although the studied problem is highly nonlinear,
when several SP data collected at different altitudes inside the column are used to calibrate the
model, all hydraulic ($K_S$, $\theta_r$, $\alpha$ and $n$) and geophysical ($n_a$ and $C_{sat}$) parameters can be
reasonably estimated from the SP measurements. Further, in this case, the FOA approach





provides accurate estimations of both mean parameter values and uncertainty regions.
Conversely, when the number of SP measurements used for the calibration is strongly
reduced, the FOA approach yields accurate mean parameter values (in agreement with
MCMC results) but inaccurate and even unphysical confidence intervals for parameters with
large uncertainty regions.

**Keywords**
Drainage experiment, Streaming Potential, Global Sensitivity Analysis, Markov chain Monte
Carlo, parameter estimation.



## 1. Introduction


Flow through a charged porous medium can generate an electric potential (Zablocki, 1978;
Ishido and Mizutani, 1981; Allegre et al., 2010; Jougnot and Linde, 2013), called Streaming
Potential (SP). The SP signals play an important role in several applications related to
hydrogeology and geothermal reservoir engineering as they are useful for examining
subsurface flow dynamics. During the last decade, surface SP anomalies have been widely
used to estimate aquifers hydraulic properties (Darnet et al., 2003). Interest on SP is
motivated by its low–cost and high sensitivity to water flow. Either coupled or uncoupled
approaches can be used for hydraulic parameter estimation from SP signals (Mboh et al.,
2012). In the uncoupled approach, Darcy velocities (e.g., Jardani et al., 2007; Bolève et al.,
2009) are obtained from tomographic inversion of SP signals and then used for the calibration
of the hydrologic model. In the coupled approach, anomalies related to the tomographic
inversion are avoided by inverting the full coupled hydrogeophysical model (Hinnell et al.,

58 2010).

The SP signals have been widely studied in saturated porous media (Bogoslovsky and Ogilvy,
1973; Patella, 1997; Sailhac and Marquis, 2001; Richards et al., 2010; Bolève et al., 2009,
among others). Fewer studies focused on the application of the SP signal in unsaturated flow
despite the big interest for such nonlinear problems (Linde et al., 2007; Allegre et al., 2010;
Mboh et al., 2012; Jougnot and Linde, 2013). Hence, in this work we are interested in the SP
signals in unsaturated porous media. Our main objective is to investigate the usefulness of the
SP signals for the characterization of soil parameters. To this aim, we evaluate the impact of
uncertain hydraulic and geophysical parameters on the SP signals and assess the identifiability
of these parameters from the SP measurements.
The impact of soil parameters on SP signals is investigated using Global Sensitivity Analysis
(GSA). This is a useful tool for characterizing the influential parameters that contribute the

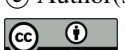



most to the variability of model outputs (Saltelli et al.,1999; Sudret, 2008) and for
understanding the behavior of the modeled system. GSA has been applied in several areas, as
for risk assessment for groundwater pollution (e.g., Volkova et al., 2008), non-reactive
(Fajraoui et al., 2011) and reactive transport experiments (Fajraoui et al., 2012; Younes et al.,
2016), for unsaturated flow experiments (Younes et al., 2013), natural convection in porous
media (Fajraoui et al., 2017) and seawater intrusion (Rajabi et al., 2015; Riva et al., 2015). To
the best of our knowledge, GSA has never been used for SP signals in unsaturated porous
media. Hence, in the first part of this study, GSA is performed on a conceptual model inspired
from the laboratory experiment of Mboh et al. (2012) where SP signals are measured at
different altitudes in a sandy soil column during a falling-head infiltration phase followed by a
drainage phase. Four uncertain hydraulic parameters (saturated hydraulic conductivity $K_S$,
residual water content $\theta_r$ and fitting Mualem-van Genuchten parameters $\alpha$ and $n$) and two
geophysical (Archies's saturation exponent $n_a$ and voltage coupling coefficient at saturation
$C_{sat}$) parameters are investigated. GSA of SP signals is performed by computing the variance-
based sensitivity indices using Polynomial Chaos Expansion (PCE). To reduce the number of
PCE coefficients while maintaining high PCE orders, we use the efficient sparse PCE
algorithm developed by Shao et al. (2017) which selects the best sparse PCE from a given
data set using the Kashyap Information Criterion (KIC).
In the second part of this study, we investigate the identifiability of hydro-geophysical
parameters from SP measurements. To this aim, parameter estimation is performed using two
different approaches. The first approach is a Bayesian approach based on the Markov Chain
Monte Carlo (MCMC) method. MCMC has been successfully used in various inverse
problems (e.g., Vrugt et al., 2003, 2008; Arora et al., 2012; Younes et al., 2017). The MCMC
method yields an ensemble of possible parameter sets that satisfactorily fit the available data.



These sets are then employed to estimate the posterior parameter distributions and hence the
optimal parameter values and the associated 95% Confidence Intervals (CIs) in order to
quantify parameter's uncertainty. The second inversion approach is the commonly used First-
Order Approximation (FOA) approach based on the standard Levenberg-Marquardt
algorithm. Besides, two scenarios are considered to investigate the effect of lack of data on
the parameter identifiability. In the first scenario, SP data collected from sensors at five
different locations are taken into account for the calibration. In the second scenario; only the
SP data from one sensor are used for model calibration.
The present study is decomposed as follows. Section 2 presents the hydrogeophysical model
and the reference solution. Section 3 reports on the GSA results of SP signals. Then, Section 4
discusses results of parameter estimation with both MCMC and FOA approaches for the two
investigated scenarios.
**2. Mathematical and conceptual models**
**2.1. Mathematical model**
The total electrical current density $j$ [A m$^{-2}$] is determined from the generalized Ohm's law
as follows:
$$j = -\sigma \nabla \varphi + j_s \tag{1}$$
where $\varphi$ [V] is the streaming potential, $j_s$ [A m$^{-2}$] is the streaming current density and $\sigma$ [S
m$^{-1}$] is the electrical conductivity distribution assumed isotropic.
Hence, the conservation equation ($\nabla . j = 0$) writes
$$\nabla . (\sigma \nabla \varphi) = \nabla . j_s \tag{2}$$
Besides, the electrical conductivity distribution can be estimated using the saturation
$S_w = \theta / \theta_s$ as follows (Mboh et al., 2012)



$$\sigma = \sigma_{sat} S_w^{\,n_a}$$
(3)

where $\sigma_{sat}$ is the electric conductivity at saturation [S m$^{-1}$] and $n_a$ is the Archies's saturation
exponent (Archie, 1942).
The streaming current density $\boldsymbol{j}_s$ can be related to the Darcy velocity $\boldsymbol{q}$ [cm min$^{-1}$] by (Linde
et al., 2007 ; Revil et al., 2007)
$$\boldsymbol{j}_s = \left( -\sigma_{sat} \frac{\rho g}{K_s} C_{sat} S_w \right) \boldsymbol{q}$$
(4)

where $K_s$ is the saturated hydraulic conductivity [cm min$^{-1}$], $\rho$ is the water density [kg m$^{-3}$],
$g$ is the gravitational acceleration [m s$^{-2}$] and $C_{sat}$ is the voltage coupling coefficient at
saturation.
Hence, the combination of the previous equations (1-4) leads to the following partial
differential equation governing the SP signals:
$$\nabla \cdot \left( S_w^{\,n_a} \nabla \varphi \right) = \nabla \cdot \left( \frac{\rho g C_{sat} S_w}{K_s} \boldsymbol{q} \right)$$
(5)

On the other hand, the flow through an unsaturated soil column can be modelled by the one-
dimensional Richard's equation:
$$\frac{\partial \theta}{\partial t} = \left( c(h) + S_s \frac{\theta}{\theta_s} \right) \frac{\partial h}{\partial t} = -\nabla \cdot \left( -K(h) \nabla (h+z) \right)$$
(6)

where $h$ [cm] is the pressure head; $z$ [cm] is the depth (downward positive); $S_s$ (-) is the
specific storage; $\theta_s$ [cm$^3$.cm$^{-3}$] and $\theta$ are the saturated and actual water contents,
respectively; $c(h)$ [cm$^{-1}$] is the specific moisture capacity; and $K(h)$ [L.T$^{-1}$] is the hydraulic
conductivity. The standard models of Mualem (1976) and Van Genuchten (1980) are used to
relate pressure head, hydraulic conductivity and water content,



$$S_e(h) = \frac{\theta(h) - \theta_r}{\theta_s - \theta_r} = \begin{cases} \dfrac{1}{\left(1 + |\alpha h|^n\right)^m} & h < 0 \\ 1 & h \geq 0 \end{cases} \tag{7}$$

$$K(S_e) = K_s S_e^{1/2} \left[ 1 - \left(1 - S_e^{1/m}\right)^m \right]^2$$

where $S_e$ (-) is the effective saturation, $\theta_r$ [L$^3$.L$^{-3}$] is the residual water content, $K_s$ [cm.min$^{-1}$] is the saturated hydraulic conductivity, $m = 1 - 1/n$, $\alpha$ [cm$^{-1}$] and $n$ [-] are the Mualem van-Genuchten shape parameters.

## 2.2. Conceptual model and numerical solution

The test case considered in this work is similar to the laboratory experiment developed in Mboh et al. (2012) involving a falling-head infiltration phase followed by a drainage phase. This experiment is representative of several laboratory SP experiments (Linde et al., 2007; Allegre et al., 2010; Jougnot and Linde, 2013, among others). Quartz sand is evenly packed in a plastic tube with an internal diameter of 5 cm to a height of $L_s = 117.5$ cm. The column is initially saturated with a ponding of $L_w = 48$ cm above the soil surface. Five sensors allowing SP measurements are installed at respectively 5, 29, 53, 77, and 101 cm from the surface. The column has a zero pressure head maintained at its bottom. At the top of the column, the boundary condition corresponds to a Dirichlet condition with a prescribed pressure head condition during the falling-head phase followed by a Neumann condition with zero infiltration flux during the drainage phase. During the falling-head phase, the prescribed pressure head $h_{top}$ has an exponential behavior driven by the saturated conductivity $h_{top} = (L_s + L_w) e^{-\frac{K_s t}{L_s}} - L_s$. The falling-head phase remains until the ponding vanishes at the critical time $t_c = -\dfrac{L_s}{K_s} \ln\left(\dfrac{L_s}{L_s + L_w}\right)$.



The sandy soil has typical MVG hydraulic parameters with (according to Mboh et al., 2012)
$K_s = 29.7$ cm/h, $\theta_s = 0.43$ cm$^3$/cm$^3$, $\theta_r = 0.045$ cm$^3$/cm$^3$, $\alpha = 0.145$ cm$^{-1}$ and $n = 2.68$. The
voltage coupling coefficient at saturation is $C_{sat} = -2.9 \, 10^{-7}$ V/Pa and the Archies's saturation
exponent is $n_a = 1.6$.
Based on these hydraulic and geophysical parameters, a reference solution is obtained using a
uniform mesh of 235 cells of 0.5 cm length. The system of equations (5)-(6) is solved with the
standard finite volume method. The temporal discretization is performed with the method of
lines (MOL) which is suitable for strongly nonlinear systems. Indeed, the MOL allows high
order temporal integration methods with formal error estimation and control (Miller et al.,
1998; Younes et al., 2009; Fahs et al., 2009, 2011).
Data are generated from the numerical model by sampling the SP signals every 10 min during
1800 min. Figure 1 shows that the SP signals have an almost linear behavior in the saturated
falling-head phase. During the drainage phase, they have a nonlinear behavior and approach
the zero voltage for the dry conditions occurring toward the end of the experiment. The SP
signals are noised with independent Gaussian random noises with a standard deviation of 2.73
$10^{-5}$ V. This noise level was obtained by Mboh et al. (2012) from laboratory measurements.
The noised data (Fig. 1) are used as "observations" in the calibration exercise.
**3. Global sensitivity analysis of SP signals**
**3.1. GSA method**
The aim of GSA is to assess the effect of the variation of parameters on the model output
(Mara and Tarantola, 2008). Such knowledge is important for determining the most influential
parameters as well as their regions and periods of influence (Fajraoui et al., 2011). The
sensitivity of a model to its parameters can be assessed using Variance-based sensitivity
indices. These indices evaluate the contribution of each parameter to the variance of the



model (Sobol', 2001). The polynomial chaos theory (Wiener, 1938), has been largely used to
perform variance-based sensitivity analysis of computer models (see for instance, Sudret,
2008; Blatman and Sudret, 2010; Fajraoui et al., 2012; Younes et al., 2016; Shao et al., 2017;
Mara et al., 2017). PCE-based sensitivity analysis is efficient since the Sobol' indices can be
directly obtained from the PCE coefficients without any additional computation (Fajraoui et
al., 2011).
Let us consider a a mathematical model with a random response $f(\xi)$ which depends on $d$
independent random parameters $\xi = \{\xi_1, \xi_2, ..., \xi_d\}$. With PCE, $f(\xi)$ is expanded using a set
of orthonormal multivariate polynomials (up to a polynomial degree $p$):

$$f(\xi) \approx \sum_{|\alpha| \le p} s_\alpha \Psi_\alpha(\xi) \qquad (8)$$

where $\alpha = \alpha_1 ... \alpha_d \in \Box^d$ is a $d^{\text{th}}$-dimensional index. The $s_\alpha$'s are the polynomial coefficients
and $\Psi_\alpha$'s are the generalized polynomial chaos of degree $|\alpha| = \sum_{i=1}^d \alpha_i$, such as Hermite,
Legendre and Jacobi polynomials, for instance. In this work, Legendre polynomials are
employed because uniform priors are considered for the parameters.
Equation (8) is similar to an ANOVA (Analysis Of Variance) representation of the original
model (Sobol' 1993), from which it is straightforward to express $V[f(\xi)]$, the variance of
$f(\xi)$ as the sum of the partial contribution of the inputs,

$$V[f(\xi)] = \sum_\alpha s_\alpha^2 , \qquad (9)$$

The first-order sensitivity index $S_i$ and the total sensitivity index $ST_i$ are defined by

$$S_i = \frac{V\left[E\left[f(\xi)|\xi_i\right]\right]}{V[f(\xi)]} \in [0,1], \qquad (10)$$





$$ST_i = \frac{E\left[V\left[f\left(\xi\right)\middle|\xi_{-i}\right]\right]}{V\left[f\left(\xi\right)\right]} \in \left[0,1\right], \tag{11}$$


where $\xi_{-i} = \xi \setminus \xi_i$, $E\left[\ \middle|\ \right]$ is the conditional expectation operator and $V\left[\ \middle|\ \right]$ the conditional
variance. $S_i$ measures the amount of variance of $f\left(\xi\right)$ due to $\xi_i$ alone, while $ST_i \geq S_i$
measures the amount of all contributions of $\xi_i$ to the variance of $f\left(\xi\right)$, including its
cooperative non-linear contributions with the other parameters $\xi_j$. The input/output
relationship is said *additive* when $ST_i = S_i$, $\forall i = 1,..,d$, and in this case $\sum_{i=1}^{d} S_i = 1$.
In the sequel, a PCE is constructed for each SP signal at each observable time. The number of
coefficients for a full PCE representation is $P = \left(d + p\right)! / \left(d! p!\right)$. The evaluation of the PCE
coefficients requires at least $P$ simulations of the nonlinear hydrogeophysical model. Note
that $P$ increases quickly with the order of the PCE and the number of parameters. Hence,
several sparse PCE representations, where only the significant coefficients are sought, have
been proposed in the literature in order to reduce the computational cost of the estimation of
the Sobol indices. For instance, Blatman and Sudret (2010) developed a sparse PCE
representation using an iterative forward-backward approach based on non-intrusive
regression. Fajraoui et al., (2012) developed a technique where only the sensitive coefficients
(that affect significantly model variance) are retained in the PCE. Recently, Shao et al.,
(2017), developed an algorithm based on Bayesian Model Averaging (BMA) to select the best
sparse PCE from a given data set using the Kashyap Information Criterion (KIC) (Kayshap,
1982). The main idea of this algorithm is to increase progressively the degree of an initial
PCE and compute the KIC until obtaining a satisfactory representation of model responses.
This algorithm is used hereafter to compute the sensitivity indices of the SP signals.



**3.2. GSA results**
The SP responses are considered for uniformly distributed parameters over the large intervals
shown in Table1. These intervals include the reference values reported in Mboh et al. (2012).
The sensitivity indices of the six input parameters $(K_s, \theta_r, \alpha, n, n_a, C_{sat})$ are estimated using an
experimental design formed by $N = 2^{12} = 4096$ parameter sets. The order of the sparse PCE is
automatically adapted for each observable time and location. For some observable times, the
PCE is highly sparse; it reaches a degree of 31 but contains only 112 nonzero coefficients.
Figure 2 depicts the temporal distribution of the streaming potential variance, represented by
the bleu curve, and the relative contribution of the parameters, represented by the shaded area.
This figure corresponds to the temporal ANOVA decomposition for the sensor 1 (at 5 cm
from the soil surface) and for the sensor 4 (at 77 cm from the soil surface). Interactions
between parameters are represented by the blank region between the variance curves and the
shaded area. Note that because Dirichlet boundary condition with zero SP is maintained at the
outlet boundary, the variance of the SP signal is zero at the bottom and reaches its maximum
value near the soil surface. Hence, the variance is higher for the first sensor, located at 5 cm
from the soil surface (Figure 2a) than for the sensor 4 located at 77 cm (Figure 2b).
The SP signals at different altitudes exhibit similar behavior (Figure 2). In the following, we
comment on the results of sensor 1 (Figure 2a). Because $K_s$ varies between 0.1 [cm min$^{-1}$]
and 2 [cm min$^{-1}$], the saturated falling-head phase remains until the ponding vanishes at
$t_c = -\dfrac{L_s}{K_s} \ln\left(\dfrac{L_s}{L_s + L_w}\right)$. Depending on the value of $K_s$ (see Table 1), $t_c$ varies between $t_1 = 20$
min and $t_2 = 403$ min. Thus, in Figure 2a, we can see that during a first time period $(t \leq t_1)$,
the SP signal is strongly influenced by the value of the parameter $C_{sat}$. The first order and
total sensitivity indices at $t = 10 \min$ (Table 2a) confirm that only the saturated parameters $K_s$





and $C_{sat}$ are influential. $C_{sat}$ is about 17 times more influential than $K_s$. As expected, the
remaining parameters have no influence during the first period. The total variance is 0.72 mv
and there is no interaction between the two parameters $K_s$ and $C_{sat}$ since $ST_i = S_i$ for both
and $\sum_{i=1}^{d} S_i = 1$.
During the second period $(t_1 \leq t \leq t_2)$, the flow is either saturated or unsaturated depending on
the value of $K_s$. Figure 2a shows that the variance of the SP signal exhibits its maximum
value around 2.4 mv with strong influences of the parameters $K_s$ and $C_{sat}$ and weak
interactions between them (small blank region between the variance curve and the shaded
area). These results are confirmed by the sensitivity indices calculated at $t = 70$ min and
reported in Table 2a for the sensor 1. Both first order and total sensitivity indices indicate that
$K_s$ is the most influential parameter. The second influential parameter is $C_{sat}$ which has a
total sensitivity index about 12 times less than $K_s$. The parameter $\alpha$ is irrelevant since its
total sensitivity index is 109 times less than $K_s$ and its partial variance is
$V_i = S_i \times V_T = 0.01 mv$ which is less than the 95% confidence interval associated to the SP
measurement ($\pm 0.055 mv$). The total variance at $t = 70$ min is calculated to be $2.17 mv$ and
the output/input relationship is close to be additive since $\sum_{i=1}^{d} S_i = 0.94$ which means that
interactions between parameters exist but are not significant.
During the third period $(t \geq t_2)$, the variance of the SP signal reduces to 0.3 mv (Figure 2a)
and significant interactions are observed between parameters (large blank region between the
shaded area and the variance curve). Table 2a shows that for $t = 800$ min, which corresponds
to dry conditions, the total variance is 0.22. First-order sensitivity indices are very small,
except for $\theta_r$. The latter is highly influential since it has a significant first-order sensitivity



index ( $S_i = 0.27$ ) and a more significant total- sensitivity index ( $ST_i = 0.74$ ). The parameters
$C_{sat}$ and $K_s$ are irrelevant, they have very small first-order and total sensitivity indices.
Further, strong interactions are observed between the parameters since the sum of the first-
order indices is far from 1 ( $\sum_{i=1}^{d} S_i = 0.47$ ). The total sensitivity indices are significantly
different from first-order sensitivity indices for almost all parameters. For instance, the ratio
between these two indices is around 4 for $\alpha$ , 5 for $n_a$ and 7 for $n$ . The total sensitivity index
of $\alpha$ remains small (0.065), whereas, significant total sensitivity indices are obtained for $n$ (
$ST_i = 0.27$ ) and $n_a$ ( $ST_i = 0.47$ ) which indicates that these two parameters are influential
(although their first order sensitivity indices are small) because of interaction between
parameters.
Figure 2b shows similar behavior for the sensor 4 located at 77 cm from the soil surface. The
results in Table 2b indicate that the total variance observed at t = 10, 70 and 800 min are
around 8 times less than for the sensor 1. For the first time period, the first and total
sensitivity indices are identical to those observed for the sensor 1 since saturated conditions
occur inside the whole column and the same effect of $K_s$ and $C_{sat}$ can be observed whatever
the location inside the column. For the second time period, the sensitivity indices for sensor 4
(Table 2b) are similar to those observed for the sensor 1. However, the results for the third
time period show an improvement of the relevance of the parameter $\alpha$ with an increase of
both first and total sensitivity indices. Indeed, compared to the results of the sensor1, both
first order and total sensitivity indices have tripled. Moreover, the total sensitivity index for $\alpha$
( $ST_i = 0.22$ ) becomes close to that of $n$ ( $ST_i = 0.24$ ).
In summary, the GSA applied to SP signals identifies the influential parameters and their
periods of influence and show that





- the parameter $C_{sat}$ is highly influential during the first time period $(t \leq t_1)$ where no
interactions are observed between parameters;
- the parameter $K_s$ is highly influential during the second time period $(t_1 \leq t \leq t_2)$ where
small interactions occur between parameters;
- the parameters $\theta_r$, $n$ and $n_a$ are influential during the third time period $(t \geq t_2)$ where
dry conditions occur. During this period, strong interactions take place between
parameters;
- the parameter $\alpha$ has no influence on the SP signals during the two first periods and
presents a very small influence ($S_i = 0.015$ and $ST_i = 0.065$) during the third period
on the sensor 1 (near the surface of the column);
- the relevance of the parameter $\alpha$ improves with the distance from the soil surface,
although the total variance diminishes with respect to this distance. The influence of
$\alpha$ becomes significant ($ST_i = 0.22$) on the sensor 4 (located at 77 cm from the soil
surface) during the third period.
**4. Parameter estimation**
**4.1. MCMC and FOA approaches**
Calibration of computer models is an essential task since some parameters (like the Mualem
van-Genuchten shape parameters $\alpha$ and $n$) cannot be directly measured. In such an exercise,
the unknown model parameters are investigated by facing the model responses to the
observations. Recently, Mboh et al. (2012) showed that inversion of SP signals can yield
accurate estimate of the saturated hydraulic conductivity $K_S$, the MVG fitting parameters $\alpha$
and $n$ and the Archie's saturation exponent ($n_a$). Moreover, they showed that the quality of
the estimation was comparable to that obtained from the calibration of pressure heads. In their



study, Mboh et al. (2012) used the FOA approach with the Shuffled Complex Evolution
optimization algorithm SCE-UA (Duan et al., 1993).
As important as the determination of the optimal parameter sets are the associated 95%
Confidence Intervals (CIs) to quantify uncertainty on the estimated values. The determination
of CIs is not straightforward if the observed model responses are highly nonlinear functions of
model parameters (Christensen and Cooley, 1999). In the sequel, parameter estimation is
performed using two approaches: the popular FOA approach and the Bayesian approach
based on the Markov chain Monte Carlo (MCMC) sampler. The MCMC method is model-
free since no assumption concerning model linearity is required for its implementation. Many
improvements have been proposed in the literature to accelerate the MCMC convergence rate
(e.g., Haario et al., 2006; ter Braak and Vrugt, 2008; Dostert et al., 2009, among others). All
MCMC samplers rely on the Metropolis-Hasting algorithm (Metropolis et al., 1953; Hastings,
1970). It proceeds as follows:


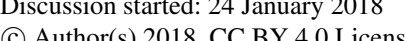


i.  Choose an initial candidate $\mathbf{x}^0 = \left(\boldsymbol{\xi}^0, \sigma^0\right)$ formed by the initial estimate of the parameter set $\boldsymbol{\xi}^0$ and the hyperparameter $\sigma^0$ and a proposal distribution $q$ that depends on the previous accepted candidate.

ii.  A new candidate $\mathbf{x}^i = \left(\boldsymbol{\xi}^i, \sigma^i\right)$ is generated from the current one $\mathbf{x}^{i-1}$ with the generator $q\left(\mathbf{x}^i \middle| \mathbf{x}^{i-1}\right)$ associated with the transition probability $p\left(\boldsymbol{\xi}^i \mid \mathbf{y}_{mes}, \sigma\right)$.

iii.  Calculate $p\left(\boldsymbol{\xi}^i \mid \mathbf{y}_{mes}, \sigma\right)$ and compute the ratio $\alpha = \dfrac{p\left(\boldsymbol{\xi}^i \mid \mathbf{y}_{mes}, \sigma\right) q\left(\mathbf{x}^i \middle| \mathbf{x}^{i-1}\right)}{p\left(\boldsymbol{\xi}^{i-1} \mid \mathbf{y}_{mes}, \sigma\right) q\left(\mathbf{x}^{i-1} \middle| \mathbf{x}^i\right)}$.

Additionally, draw a random number $u \in [0,1]$ from a uniform distribution.

iv.  If $\alpha \geq u$, then accept the new candidate, otherwise it is rejected.

v.  Resume from (ii) until the chain $\left\{\mathbf{x}^0, ..., \mathbf{x}^k\right\}$ converges or a prescribed number of iterations $i_{max}$ is reached.

Recently, Laloy and Vrugt (2012) developed the DREAM$_{(ZS)}$ MCMC sampler which runs multiple chains in parallel for a wider and quicker exploration of the parameter space. However, because of the large number of model evaluations required, the MCMC method remains rarely used compared to the FOA approach. Indeed, with FOA, the CIs are estimated once by assuming that the Jacobian remains constant within the CIs. This assumption was found to be reasonably accurate in nonlinear problems by Donaldson and Scnabel (1987). However, recently, several authors stated that parameter interdependences and model nonlinearities violate this assumption (see for instance, Vrugt and Bouten, 2002; Vurgin et al. 2007; Gallagher and Doherty, 2007; Mertens et al., 2009; Kahl et al., 2015).

In the following, both MCMC and FOA approaches are employed for the inversion of the highly nonlinear hydrogeophysical problem using SP measurements.





### 4.2. Parameter estimation results

Hydrogeophysical parameters are estimated using the DREAM$_{(ZS)}$ MCMC sampler (Laloy
and Vrugt, 2012). Independent uniform distributions are considered for model parameter
priors and likelihood hyperparameters (see Table 1). The parameter posterior distribution
writes:

$$p\left(\boldsymbol{\xi}/\boldsymbol{y}_{mes},\sigma\right) \propto \sigma^{-N} exp\left(-\frac{SS\left(\boldsymbol{\xi}\right)}{2\sigma^{2}}\right) \qquad (9)$$

where $SS\left(\boldsymbol{\xi}\right)=\sum_{k=1}^{N}\left(y_{mes}^{(k)}-y_{mod}^{(k)}\left(\boldsymbol{\xi}\right)\right)^{2}$ is the sum of the squared differences between the
observed $y_{mes}^{(k)}$ and modeled $y_{mod}^{(k)}$ SP signals at time $t_{k}$ for $N$ total number of SP
observations.
The DREAM$_{(ZS)}$ software computes multiple sub-chains in parallel to thoroughly explore the
parameter space. Taking the last 25% of individuals (when the chains have converged) yields
multiple sets used to estimate the updated parameter distributions and therefore the optimal
parameter values and their CIs. In the sequel, the DREAM$_{(ZS)}$ MCMC sampler is used with 3
parallel chains.
We assume that the saturated water content has been initially measured with a fair degree of
accuracy. However, instead of fixing its value (as in Kool et al. (1987), van Dam et al. (1994),
Nützmann et al., (1998) among others), we assign to $\theta_{s}$ a Gaussian distribution to take into
account associated uncertainty and its effect on the estimation of the rest of parameters. Hence
a Gaussian distribution is assigned to $\theta_{s}$ with a mean value of 0.43 cm$^{3}$.cm$^{-3}$ and a 95% CI
$\left[0.41-0.45\right]$ cm$^{3}$.cm$^{-3}$. The rest of parameters are uniformly distributed over the ranges
reported in Table 1. The standard deviation $\sigma$ is also considered unknown and is
simultaneously estimated with the physical parameters. Two scenarios are considered: in the
first scenario, SP data collected from the sensors located at the five locations are taken into





account for the calibration. In the second scenario; only the SP data from the first sensor
located at 5 cm from the soil surface serve as conditioning information for model calibration.
Results of the MCMC sampler are compared to those of FOA approach for both scenarios.

**3.1 Scenario 1: Inversion using all SP measurements**

Fig. 3 shows the results obtained with MCMC when the SP data of the five sensors are used
for the calibration. The "on-diagonal" plots in this figure display the posterior parameter
distributions, whereas the "off-diagonal" plots represent the correlations between parameters
in the MCMC sample. Fig. 3 shows bell-shaped posterior distributions for all parameters. A
strong correlation is observed between $\theta_r$ and $n_a$ ( $r = 0.98$ ).
From the obtained MCMC sample, it is straightforward to estimate the posterior 95%
confidence interval of each parameter. The latter as well as the mean estimate value of each
parameter obtained with both MCMC and FOA approaches are reported in Table 3.
The results this table show that the parameters are well estimated from the SP measurements
since (*i*) identified mean values are very close to the reference solution, (*ii*) all confidence
intervals include the reference solution and (*iii*) the confidence intervals are rather narrow.
The saturated parameters $K_S$ and $C_{sat}$ are very well estimated (with CIs around 2%) because
of data collected during the falling-head phase where only these two parameters are
influential.
The posterior CI of the parameter $\theta_s$ is similar to its prior CI. The parameter $\alpha$ is reasonably
well estimated with a CI around 35%. Recall that this parameter had very small first-order and
total sensitivity indices for sensor 1 but had more significant sensitivity indices for the sensors
away from the soil surface (see results for sensor 4 in Table 2b). The parameter $\theta_r$ is
estimated with a CI around 90% although it was highly influential for all sensors (for
instance, a first-order sensitivity index of 0.27 and a total order of 0.74 for sensor 1). The





parameters $n$ and $n_a$ had similar GSA behavior with small first-order sensitivities
(respectively 0.038 and 0.094 for sensor 1) and large total sensitivities (respectively 0.266 and
0.4715 for sensor 1), however, the inversion shows that the parameter $n$ is well estimated
with a CI less than 10% whereas the parameter $n_a$ is less well estimated with a CI around
35%. These results suggest that GSA outcomes should be interpreted with caution in the
context of parameter estimation since (*i*) a parameter which is not relevant for the model
output in one sensor can be influential for another sensor and (*ii*) GSA does not presume on
the quality of the estimation since two parameters with similar sensitivity indices can have
different quality of estimation by the inversion procedure.
Further, the results of Table 3 show that FOA and MCMC approaches yield similar mean
estimated values. Moreover, very good agreement is observed between FOA and MCMC
uncertainty bounds. Concerning the efficiency of the two calibration methods for this
scenario, the FOA approach is by far the most efficient method since it requires only 95s of
CPU time. The MCMC method was terminated after 15,000 model runs which required
14,116s. The convergence was reached at around 10,000 model runs. The last 5,000 runs were
used to estimate the statistical measures of the posterior distribution.
**3.2 Scenario 2: Inversion using only SP measurements near the surface**
In this scenario, the number of measurements used for the calibration is strongly reduced.
Only SP measurements from sensor 1 (located at 5 cm blow de soil surface) are considered.
The results of MCMC are plotted in the Fig. 4. The correlation observed between $\theta_r$ and $n_a$
decreases slightly to $r = 0.95$. Almost bell-shaped posterior distributions are observed for all
parameters except for the parameters $\theta_r$ and $\alpha$.
The results obtained with MCMC and FOA approaches depicted in Table 4 show that





-    The FOA approach yields accurate mean estimated values similar to MCMC results

for all parameters;

-    The MCMC and FOA mean estimated values are close to the reference solution and to

the previous scenario. The maximum difference is observed for $\theta_r$ for which the

mean estimated value with scenario 2 is 15% greater than for scenario 1

-    The MCMC CIs for the parameters $K_S$, $\theta_s$, $n$ and $C_{sat}$ are close to the previous

scenario. The parameters $\theta_s$ and $n$ are well estimated (CIs < 10%) and the

parameters $K_S$ and $C_{sat}$ are very well estimated (CIs ≤ 5%).

-    Due to the reduction of the number of data used for model calibration in the scenario

2, the MCMC CIs for the parameters $n_a$, $\alpha$ and $\theta_r$ are much larger than in the

previous scenario. Indeed, compared to scenario1, the CI for $n_a$ and $\theta_r$ increases by

around 60% whereas the CI of $\alpha$ is 3 times larger than for the scenario 1.

-    The FOA method yields accurate CIs for the parameters $\theta_s$, $n$, $n_a$ and $C_{sat}$ whereas it

overestimates the CIs of $\theta_r$ (by 24%), $K_S$ (by 100%) and $\alpha$ (by 427%). Unphysical

uncertainty region (including negative values) is obtained for the parameter $\alpha$

These results show that the FOA can fail to provide realistic parameter uncertainties and can
yield larger CIs than their corresponding nonlinear MCMC counterpart. Indeed, the
linearization in the FOA method assumes that the Jacobian remains constant across the CIs.
This assumption was quite fulfilled for the first scenario in which a large number of
measurements insured small uncertainty regions. However, the assumption is not fulfilled for
some parameters of the current scenario because of the large uncertainty regions induced by
the reduction of the number of SP measurements.
Concerning the efficiency of the calibration methods, the FOA required approximately 174s
of CPU time, the MCMC required much more runs to reach the convergence than in the





previous scenario. Indeed, the sampler was used with 50,000 runs (35,000 runs were
necessary to reach the convergence).

## 4. Conclusions

In this work, a synthetic test case dealing with SP signals during drainage experiment has
been studied. The test case is similar to the laboratory experiment developed in Mboh et al.
(2012) involving a falling-head infiltration phase followed by a drainage phase. GSA and
Bayesian parameter inference have been applied to investigate (*i*) the influence of hydraulic
and geophysical parameters on the SP signals and (*ii*) the identifiability of hydro-geophysical
parameters using only SP measurements. The GSA was performed using variance-based
sensitivity indices which allow measuring the contribution of each parameter (alone or by
interaction with other parameters) to the output variance. The sensitivity indices have been
calculated using a PCE representation of the SP signals. To reduce the number of coefficients
and explore PCE with high orders, we used the efficient sparse PCE algorithm developed by
Shao et al. (2017) which selects the best sparse PCE from a given data set using the Kashyap
Information Criterion (KIC).
The GSA applied to SP signals showed that the parameters $C_{sat}$ and $K_s$ are highly influential
during the first period corresponding to saturated conditions. The parameters $\theta_r$, $n$ and $n_a$
are influential when dry conditions occur. In such conditions, strong interactions take place
between these parameters. The parameter $\alpha$ has a very small influence on the SP signals near
the soil surface but its sensitivity increases with depth although the total variance decreases
with depth.
Parameter estimation has been performed using MCMC and FOA approaches. All hydraulic (
$K_S$, $\theta_r$, $\alpha$ and $n$ ) and geophysical ($n_a$ and $C_{sat}$) parameters can be reasonably estimated in
the first scenario when the whole SP data (measured at five different locations) are used as



conditioning information for the model calibration. The confrontation with GSA results shows
that the latter should be interpreted with caution when used in the context of parameter
estimation since (*i*) a parameter which is not relevant for the model output in one sensor can
be influential for another sensor and (*ii*) GSA does not presume on the quality of the
estimation since two parameters with similar sensitivity indices can have different quality of
estimation by the inverse procedure (see for instance, parameters $n$ and $n_a$). Furthermore,
although the studied problem is highly nonlinear, the FOA approach provides accurate
estimations of both mean parameter values and CIs in the first scenario and is by far much
more efficient than the MCMC method.
When the number of SP measurements used for the calibration is considerably reduced (lack
of data), the MCMC inversion provides larger parameters' uncertainty regions. The FOA
approach yields accurate mean parameter values (in agreement with MCMC results) but
inaccurate and even unphysical CIs for some parameters with large uncertainty regions.





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





**List of table captions**
Table 1. Reference values, lower and upper bounds for hydraulic and geophysical parameters.
Table 2. The first-order sensitivity index $S_i$ and the total sensitivity index $ST_i$ for the SP
signal at 5 cm and 77 cm below the soil surface at different times.
Table 3: Estimated mean values (underlined), confidence intervals (CIs) and size of the
posterior CIs (italic) with MCMC and FOA approaches for scenario 1.
Table 4: Estimated mean values (underlined), confidence intervals (CIs) and size of the
posterior CIs (italic) with MCMC and FOA approaches for scenario 2.



| Parameters | Lower bounds | Upper bounds | Reference values |
|---|---|---|---|
| $K_s$ [cm min$^{-1}$] | 0.1 | 2 | 0.495 |
| $\theta_r$ [cm$^3$ min$^{-3}$] | 0 | 0.2 | 0.045 |
| $\alpha$ [cm$^{-1}$] | 0.01 | 0.2 | 0.145 |
| $n$ | 1.5 | 7 | 2.68 |
| $n_a$ [-] | 1 | 3 | 1.6 |
| $C_{sat} \times \left(-10^{-7}\right)$ [V/Pa] | 2 | 4 | 2.9 |

Table 1. Reference values, lower and upper bounds for hydraulic and geophysical parameters.

| | $K_s$ | $\theta_r$ | $\alpha$ | $n$ | $n_a$ | $C_{sat}$ |
|---|---|---|---|---|---|---|
| **a- sensor 1 (5 cm from the soil surface)** | | | | | | |
| t=10 min (total variance = 0.72) | | | | | | |
| $S_i$ | 0.055 | 0 | 0 | 0 | 0 | 0.942 |
| $ST_i$ | 0.057 | 0 | 0 | 0 | 0 | 0.945 |
| t=70 min (total variance = 2.17) | | | | | | |
| $S_i$ | 0.841 | 0.217 | 0.005 | 0.014 | 0.008 | 0.045 |
| $ST_i$ | 0.894 | 0.043 | 0.008 | 0.028 | 0.021 | 0.078 |
| t=800 min (total variance = 0.224) | | | | | | |
| $S_i$ | 0.053 | 0.266 | 0.015 | 0.038 | 0.094 | 0.008 |
| $ST_i$ | 0.085 | 0.738 | 0.065 | 0.266 | 0.472 | 0.041 |
| **b- sensor 4 (77 cm from the soil surface)** | | | | | | |
| t=10 min (total variance = 0.094) | | | | | | |
| $S_i$ | 0.055 | 0 | 0 | 0 | 0 | 0.942 |
| $ST_i$ | 0.057 | 0 | 0 | 0 | 0 | 0.945 |
| t=70 min (total variance = 0.2744) | | | | | | |
| $S_i$ | 0.839 | 0.015 | 0.014 | 0.013 | 0.005 | 0.053 |
| $ST_i$ | 0.891 | 0.028 | 0.024 | 0.025 | 0.011 | 0.086 |
| t=800 min (total variance = 0.224) | | | | | | |
| $S_i$ | 0.099 | 0.225 | 0.054 | 0.043 | 0.085 | 0.01 |
| $ST_i$ | 0.138 | 0.621 | 0.218 | 0.238 | 0.379 | 0.043 |

Table 2. The first-order sensitivity index $S_i$ and the total sensitivity index $ST_i$ for the SP

signal at 5 cm and 77 cm below the soil surface at different times.

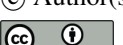



|  | MCMC | FOA |
|---|---|---|
| $K_s$ | **0.49** <br> (0.487-0.498) <br> *0.01* | **0.49** <br> (0.487-0.497) <br> *0.01* |
| $\theta_s$ | **0.43** <br> (0.41-0.45) <br> *0.04* | **0.43** <br> (0.41-0.45) <br> *0.04* |
| $\theta_r$ | **0.046** <br> (0.025-0.068) <br> *0.04* | **0.046** <br> (0.026-0.066) <br> *0.04* |
| $\alpha$ | **0.14** <br> (0.12-0.17) <br> *0.05* | **0.14** <br> (0.12-0.16) <br> *0.04* |
| $n$ | **2.64** <br> (2.54-2.77) <br> *0.23* | **2.64** <br> (2.54-2.76) <br> *0.22* |
| $n_a$ | **1.64** <br> (1.37-1.98) <br> *0.6* | **1.64** <br> (1.38-1.90) <br> *0.5* |
| $C_{sat}$ | **2.90** <br> (2.89-2.91) <br> *0.02* | **2.90** <br> (2.89-2.91) <br> *0.02* |

Table 3: Estimated mean values (underlined), confidence intervals (CIs) and size of the

posterior CIs (italic) with MCMC and FOA approaches for scenario 1.





|         | MCMC            | FOA             |
|---------|-----------------|-----------------|
| $K_S$   | **0.49**        | **0.49**        |
|         | (0.481-0.495)   | (0.474-0.503)   |
|         | *0.014*         | *0.029*         |
| $\theta_s$ | **0.43**     | **0.43**        |
|         | (0.41-0.45)     | (0.41-0.45)     |
|         | *0.04*          | *0.04*          |
| $\theta_r$ | **0.053**    | **0.053**       |
|         | (0.011-0.093)   | (0.002-0.103)   |
|         | *0.08*          | *0.1*           |
| $\alpha$ | **0.13**       | **0.13**        |
|         | (0.07-0.20)     | (-0.15-0.43)    |
|         | *0.13*          | *0.58*          |
| $n$     | **2.54**        | **2.56**        |
|         | (2.44-2.68)     | (2.44-2.68)     |
|         | *0.24*          | *0.24*          |
| $n_a$   | **1.82**        | **1.78**        |
|         | (1.36-2.41)     | (1.29-2.27)     |
|         | *1.05*          | *0.98*          |
| $C_{sat}$ | **2.89**      | **2.89**        |
|         | (2.88-2.91)     | (2.88-2.91)     |
|         | *0.03*          | *0.03*          |

Table 4: Estimated mean values (underlined), confidence intervals (CIs) and size of the

posterior CIs (italic) with MCMC and FOA approaches for scenario 2.





488              **List of figure captions**

Fig. 1. Reference SP signals. Solid lines represent the reference SP solution and dots represent
the sets of perturbed data serving as conditioning information for model calibration.

Figure 2. Time distribution of the SP variance at 5cm (a) and 77cm (b) depth. The shaded area
under the variance curve represents the partial marginal contributions of the random input
parameters; the contribution of interactions between parameters is represented by the blank
region between the shaded area and the variance curve.

Fig. 3: MCMC solutions when all SP data are considered for the calibration. The diagonal
plots represent the inferred posterior probability distribution of the model parameters. The
off-diagonal scatterplots represent the pairwise correlations in the MCMC drawing.

Fig. 4: MCMC solutions when calibration is performed using only SP data located at 5 cm
from the surface. The diagonal plots represent the posterior probability distribution of the
parameters. The off-diagonal scatterplots represent the pairwise correlations in the MCMC
drawing.





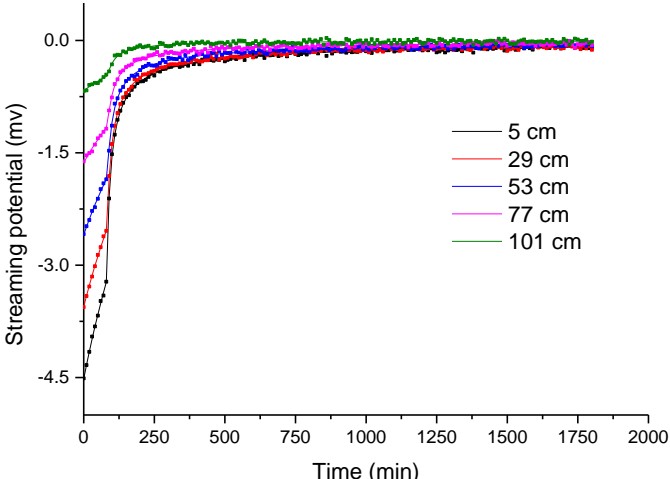

Fig. 1. Reference SP signals. Solid lines represent the reference SP solution and dots represent

the sets of perturbed data serving as conditioning information for model calibration.




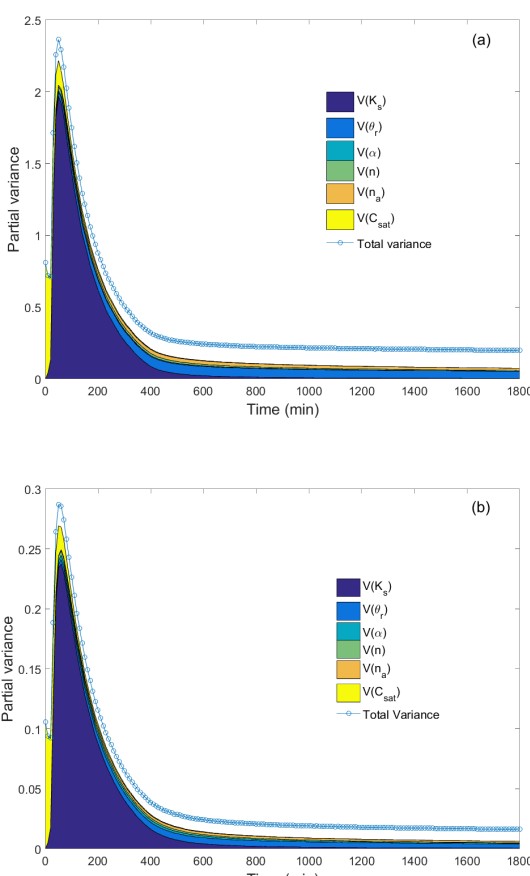

Figure 2. Time distribution of the SP variance at 5cm (a) and 77cm (b) depth. The shaded area

under the variance curve represents the partial marginal contributions of the random input

parameters; the contribution of interactions between parameters is represented by the blank

region between the shaded area and the variance curve.





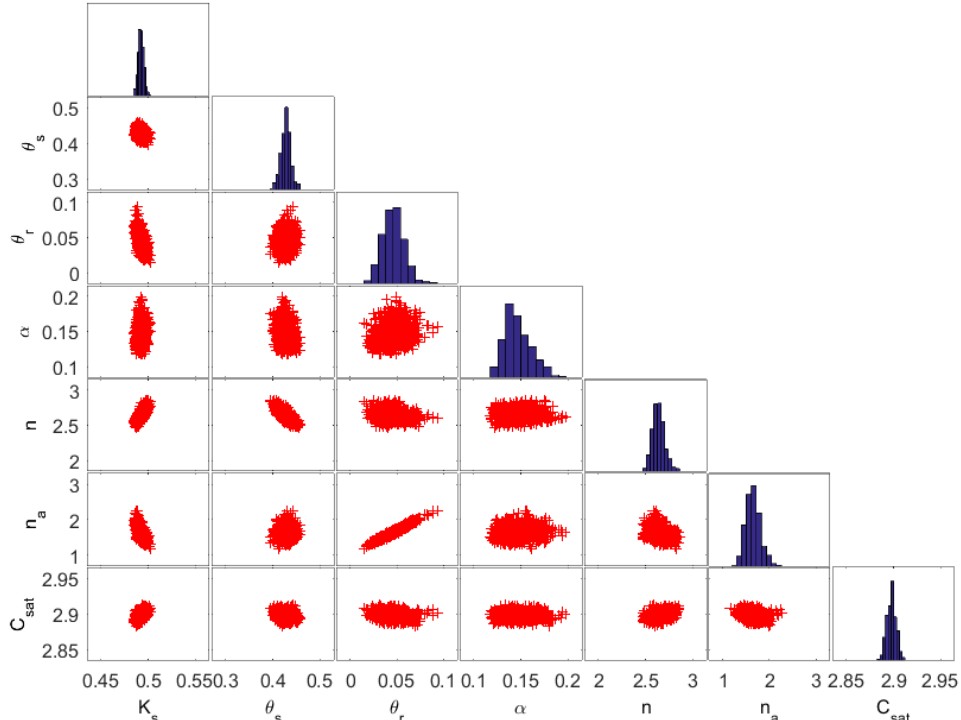

Fig. 3: MCMC solutions when all SP data are considered for the calibration. The diagonal

plots represent the inferred posterior probability distribution of the model parameters. The

off-diagonal scatterplots represent the pairwise correlations in the MCMC drawing.






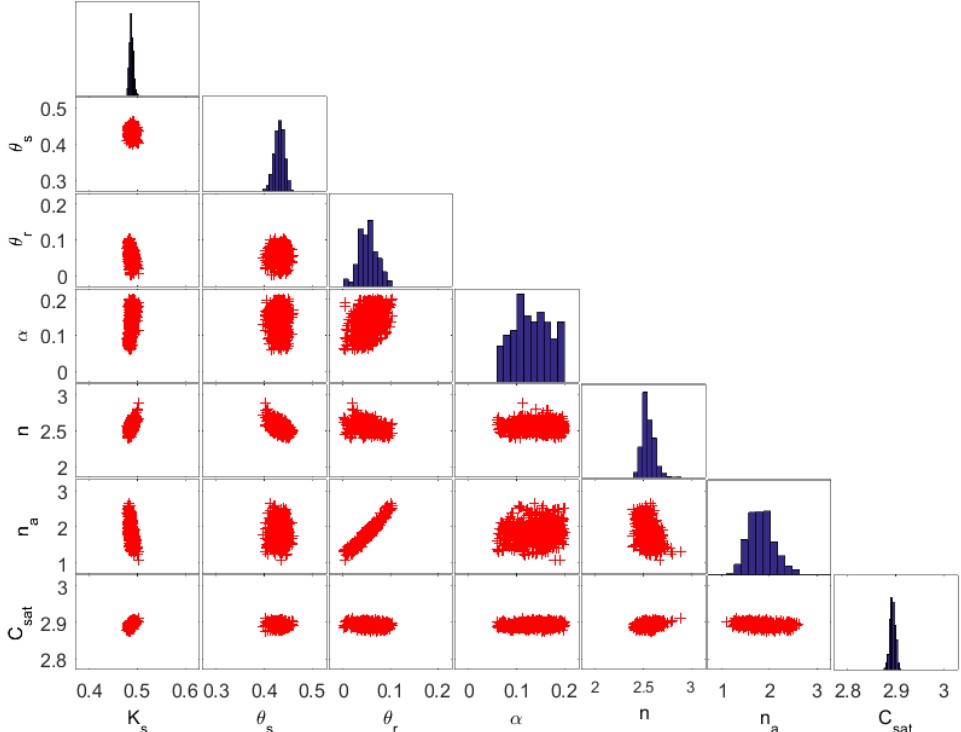

Fig. 4: MCMC solutions when calibration is performed using only SP data located at 5 cm

from the surface. The diagonal plots represent the posterior probability distribution of the

parameters. The off-diagonal scatterplots represent the pairwise correlations in the MCMC

drawing.