# Peer review of "Sensitivity and identifiability of hydraulic and geophysical parameters from streaming potential signals in unsaturated porous media"

_Hydrology and Earth System Sciences, 2017_

## Referee Comment (RC1) · Anonymous Referee #1 · 8 Feb 2018

The authors performed modelling of fluid flow in a charged porous media. They used Global Sensitivity Analysis (GSA) and parameter estimation to assess the effect of hydraulic and geophysical parameters on the streaming potentials. The subject is interesting, important and useful and deserves to be published. However, there are still some key points need to be addressed. This reviewer recommends to do some revision taking into account the below comments. 1. A section should be added for the numerical how to solve Eqs. (1-7), such as grid strategy, discrete method and converge criteria. If use a commercial software, the software needs to be cited. 2. Line 141, this section is the test case, and therefore, this should not be called conceptual model. Furthermore, a schematic of the test case should be added to show

main dimensions and boundaries 3. Line 162, what is "the standard finite volume method"? Finite volume method is a big family to solve partial differential equations, such as a first-order and second-order approximation/discretion. 4. Although the authors have performed a good review of literature of streaming potential signals, other literatures of unsaturated porous media should be introduced, such as Deng and Wang, Saturated-unsaturated groundwater modelling using 3D Richards equation with a co-ordinate transform of nonorthogonal grids, Applied Mathematical Modelling 2017, 50: 39–52.

---

## Referee Comment (RC2) · Anonymous Referee #2 · 18 Feb 2018

The paper addresses the important topic of assessing the influence of hydraulic and geophysical parameters on streaming potential in the unsaturated zone. I think that the topic would be interesting to the journals audience. Furthermore the paper is generally well written and includes novel results. My main concerns and recommendation are as follows: (1) What is the notion behind replacing the numerical model with data-driven PCE in global sensitivity analysis? I mean PCE is often used when the computational cost is prohibitive, and off course, at the cost of reduced accuracy. However, the computational cost does not seem to be high in the current study. So justification is needed on this issue. (2) Why does the study use two different method for parameter estimation? Is the objective (a) comparison of the two methods, or (b) double checking

the results (or possibly both)? I suggest that the authors clearly define the objective. Moreover, since the two methods are conceptually very different (e.g. sampling vs. optimization), if the objective includes (a), then attention must be given to other issues such as the very different and superior information content of the results in MCMC, and its ability to effectively employ prior information. (3) MCMC is not a method but a general class of strategies that may not necessarily be based on the steps described in page 17. Hence, I suggest that before describing the steps, the authors mention the specific algorithm used in the study (i.e. DREAM) and then focus their discussions on how DREAM works. Please also add reference for the MCMC steps. (4) Some other choices in the paper also require further explanations and justification. These include: (a) choice of the uniform priors (line 193), (b) choice of parameters of the Gaussian distribution for teta (lines 366-367), and (c) the logic behind the definition of scenarios in section 4.2.

Minor Comments: (1) The numbering of some sections requires correction (e.g. after section 4.2 we have section 3.1) (2) Line 83: Move "parameters" before the parenthesis. (3) Section 2.2: I also suggest adding a schematic figure and possible a photo of the laboratory setup. (4) Line 166: Data are generated "from" or "for"? (5) Line 186: Delete additional "a". (6) Line 193: Why use the word "prior" and not simply "distribution", since the authors are discussing GSA in section 3.1, and not a Bayesian inference method. (7) Line 230: Correct typing error for "blue". (8) Line 289: Correct typo to "shows that:" (9) Line 357: Add reference for the DREAM software. (10) Line 378: Not all sub-plots of figure 3 are symmetric though (e.g. plot for alpha) and so some cannot be considered bell-shaped. (11) Fig. 2: The colors are not so distinct (e.g. blue and blue-green) and so the difference between the plots may be hard to conceive for some readers. (12) Fig 3 and 4: Purely as a suggestion, the histograms can be replaced with PDFs, as I think this would provide better visualization.

---

## Author Comment (AC1) · 10 Apr 2018

We thank the reviewer for his/her detailed comments that helped us clarify the manuscript and avoid misinterpretations.

The authors performed modelling of fluid flow in a charged porous media. They used Global Sensitivity Analysis (GSA) and parameter estimation to assess the effect of hydraulic and geophysical parameters on the streaming potentials. The subject is interesting, important and useful and deserves to be published. We thank the reviewer for his/her positive appraisal of the subject of our work. However, there are still some key points need to be addressed. This reviewer recommends to do some revision taking into account the below comments.

1. A section should be added for the numerical how to solve Eqs. (1-7), such as grid strategy, discrete method and converge criteria. If use a commercial software, the software needs to be cited. We agree and add a new section describing the numerical solution of the Eqs (1-7). The solution is based on the Finite volume method coupled to a higher time integration scheme. The temporal discretization of the obtained non-linear ODE/DAE system is performed with the method of lines (MOL) using the DASPK (Brown et al., 1994) time solver. The MOL is suitable for strongly nonlinear systems since it allows high order temporal integration methods with formal error estimation and control (Miller et al., 1998; Younes et al., 2009; Fahs et al., 2009, 2011). In the current study, the relative and absolute local error tolerances are fixed to 10-6. A mesh sensitivity analysis is also performed to ensure a mesh independent solution.

2. Line 141, this section is the test case, and therefore, this should not be called conceptual model. Furthermore, a schematic of the test case should be added to show main dimensions and boundaries

We agree, change the title of the section and add a schematic figure of the test case.

3. Line 162, what is "the standard finite volume method"? Finite volume method is a big family to solve partial differential equations, such as a first-order and second-order approximation/discretion. This sentence is removed from the revised version since a new section is added dealing with the finite volume discretization of the system of equations (see comment 1).

4. Although the authors have performed a good review of literature of streaming potential signals, other literatures of unsaturated porous media should be introduced, such as Deng and Wang, Saturated-unsaturated groundwater modelling using 3D Richards equation with a coordinate transform of nonorthogonal grids, Applied Mathematical Modelling 2017, 50: 39–52. We agree and add new references related to numerical solution of Richards equation in the revised version.

Please find attached the revised version of the paper.

Please also note the supplement to this comment:
https://www.hydrol-earth-syst-sci-discuss.net/hess-2017-730/hess-2017-730-AC1-
supplement.pdf

—————————————————
730, 2018.

[Figure]

**Supplement:**

[revised manuscript text omitted]

---

## Author Comment (AC2) · 10 Apr 2018

We thank the reviewer for his/her thoughtful and detailed comments that definitely helped us clarify the manuscript and avoid misinterpretations.

The paper addresses the important topic of assessing the influence of hydraulic and geophysical parameters on streaming potential in the unsaturated zone. I think that the topic would be interesting to the journals audience. Furthermore the paper is generally well written and includes novel results. We thank the reviewer for his/her positive appraisal of our work.

Comments:

(1) What is the notion behind replacing the numerical model with data-driven PCE in global sensitivity analysis? I mean PCE is often used when the computational cost is prohibitive, and off course, at the cost of reduced accuracy. However, the computational cost does not seem to be high in the current study. So justification is needed on this issue. The Sobol' indices are introduced in the framework of ANalysis Of VAriance (ANOVA) which aims at decomposing the variance of the output as a sum of contributions of each input variable, or combinations between them. It can be stated that the PCE method is a surrogate-based approach. However, we argue that this method employs ANOVA-like decomposition and hence can be considered as a spectral method (such as the Fourier amplitude sensitivity test, Cukier et al., 1973; Saltelli et al., 1999). Indeed, the Sobol' indices are directly obtained from the PCE coefficients without needing to run the surrogate model. This point is specified in the revised version (2) Why does the study use two different methods for parameter estimation? Is the objective (a) comparison of the two methods, or (b) double checking the results (or possibly both)? I suggest that the authors clearly define the objective. Moreover, since the two methods are conceptually very different (e.g. sampling vs. optimization), if the objective includes (a), then attention must be given to other issues such as the very different and superior information content of the results in MCMC, and its ability to effectively employ prior information. According to this comment, we specify in the revised version that (i) FOA and MCMC are conceptually very different and contrarily to FOA, the MCMC method is robust since no assumptions of model linearity or differentiability are required. Furthermore, MCMC can include prior information available for the parameters and yields not only the optimal point estimate of the parameters but also a quantification of the entire parameter space. (ii) The comparison between FOA and MCMC methods is performed to check whether the popular FOA approach can provide reliable estimation of parameters and associated uncertainties for the investigated highly nonlinear hydro-geophysical problem both in the case of abundant data (small uncertainty regions) and in the case of scarcity of data (large uncertainty regions).

(3) MCMC is not a method but a general class of strategies that may not necessarily be based on the steps described in page 17. Hence, I suggest that before describing the steps, the authors mention the specific algorithm used in the study (i.e. DREAM) and then focus their discussions on how DREAM works. Please also add reference for the MCMC steps. We agree and according to this comment (i) we specify before describing the steps that several MCMC strategies have been developed for Bayesian sampling of the parameter space (Gallagher and Doherty, 2007; Vrugt, 2016). In groundwater and vadose zone modeling context, the most widely used of these strategies is the Metropolis Hastings algorithm (Metropolis et al., 1953; Hastings, 1970) described in the paper. (ii) We add the reference Gelman et al. (1996) for the Metropolis Hastings algorithm steps (iii) More details are provided concerning DREAM and DREAM(ZS). Indeed, in the revised version, we specify that Vrugt et al. (2009b, 2009c) developed the DREAM MCMC sampler based on the differential evolution–Markov Chain method of ter Braak (2006) to improve sampling efficiency. DREAM runs multiple Markov chains in parallel and uses subspace sampling and outlier chain correction to speed up MCMC convergence (Vrugt, 2016). Laloy and Vrugt (2012) developed the DREAM(ZS) MCMC sampler in which a candidate for each chain is drawn from an archive of past states denoted Z which plays the role of the generator q. The interested readers are referred to Vrugt (2016) for more details about properties and implementation of DREAM and DREAM(ZS). In the current study, the DREAM(ZS) software is used for the MCMC estimation of the hydrogeophysical parameters. These points are specified in the revised version (4) Some other choices in the paper also require further explanations and justification. These include: (a) choice of the uniform priors (line 193), (b) choice of parameters of the Gaussian distribution for teta (lines 366-367), and (c) the logic behind the definition of scenarios in section 4.2. We agree and specify in the revised version that (i) The non-informative uniform distributions are used here to express the absence of prior information which makes all possible values of the parameter equally likely. (ii) The saturated water content is assumed accurately measured by weighing the saturated soil to be tetas=0.43cm3.cm-3. The error measurements are assumed to be

independently and normally distributed with a zero mean and a standard deviation 0.01 cm3.cm-3. (iii) The two scenarios are considered to check whether the FOA approach can provide reliable estimation of parameters and associated uncertainties for the investigated highly nonlinear hydrogeophysical problem both in the case of abundant data (small uncertainty regions) and in the case of scarcity of data (large uncertainty regions).

Minor Comments: All the comments have been taken into account in the revised version

(1) The numbering of some sections requires correction (e.g. after section 4.2 we have section 3.1) We agree and correct the numbering in the revised version (2) Line 83: Move "parameters" before the parenthesis. Done in the revised version (3) Section 2.2: I also suggest adding a schematic figure and possible a photo of the laboratory setup. A schematic figure with the experimental device has been added.

(4) Line 166: Data are generated "from" or "for"? laboratory setup. The sentence has been changed.

(5) Line 186: Delete additional "a". Done in the revised version

(6) Line 193: Why use the word "prior" and not simply "distribution", since the authors are discussing GSA in section 3.1, and not a Bayesian inference method. We agree and change the word "prior" by "distribution" in the revised version

(7) Line 230: Correct typing error for "blue". Done in the revised version

(8) Line 289: Correct typo to "shows that:" Done in the revised version (9) Line 357: Add reference for the DREAM software. Done in the revised version (10) Line 378: Not all sub-plots of figure 3 are symmetric though (e.g. plot for alpha) and so some cannot be considered bell-shaped. We agree and change the sentence to nearly bell-shaped

(11) Fig. 2: The colors are not so distinct (e.g. blue and blue-green) and so the difference between the plots may be hard to conceive for some readers. Done in the

revised version

(12) Fig 3 and 4: Purely as a suggestion, the histograms can be replaced with PDFs, as I think this would provide better visualization. We obtain a better visualization with histograms that we keep in the Figures 3 and 4.

Please find attached the revised version of the paper.

Please also note the supplement to this comment:
https://www.hydrol-earth-syst-sci-discuss.net/hess-2017-730/hess-2017-730-AC2-supplement.pdf